# Artificial Intelligence-Based Management of Adult Chronic Myeloid Leukemia: Where Are We and Where Are We Going?

**DOI:** 10.3390/cancers16050848

**Published:** 2024-02-20

**Authors:** Simona Bernardi, Mauro Vallati, Roberto Gatta

**Affiliations:** 1Department of Clinical and Experimental Sciences, University of Brescia, 25123 Brescia, Italy; simona.bernardi@unibs.it; 2CREA—Centro di Ricerca Emato-Oncologica AIL, ASST Spedali Civili of Brescia, 25123 Brescia, Italy; 3School of Computing and Engineering, University of Huddersfield, Huddersfield HD1 3DH, UK; m.vallati@hud.ac.uk

**Keywords:** chronic myeloid leukemia, artificial intelligence, machine learning, minimal residual disease, risk assessment

## Abstract

**Simple Summary:**

The field of artificial intelligence (AI) is quickly becoming recognized for its potential to significantly improve medicine. AI is still in its infancy when it comes to treating Chronic Myeloid Leukemia (CML), which was once thought to be an easily treated cancer until TKIs were introduced and significantly increased patient survival. Notably, preliminary trial results are intriguing and promising in terms of AI’s performance and flexibility to be used in many scenarios. With a general focus that extends beyond Machine Learning (ML) and embraces the broader AI area, in this review we describe the state of the art of AI applications in the field of CML, including the methods and goals. We also take advantage of the occasion to talk about the primary dangers and crucial issues that AI needs to address, particularly in light of the crucial role that the “human” element plays and how important it is in this field.

**Abstract:**

Artificial intelligence (AI) is emerging as a discipline capable of providing significant added value in Medicine, in particular in radiomic, imaging analysis, big dataset analysis, and also for generating virtual cohort of patients. However, in coping with chronic myeloid leukemia (CML), considered an easily managed malignancy after the introduction of TKIs which strongly improved the life expectancy of patients, AI is still in its infancy. Noteworthy, the findings of initial trials are intriguing and encouraging, both in terms of performance and adaptability to different contexts in which AI can be applied. Indeed, the improvement of diagnosis and prognosis by leveraging biochemical, biomolecular, imaging, and clinical data can be crucial for the implementation of the personalized medicine paradigm or the streamlining of procedures and services. In this review, we present the state of the art of AI applications in the field of CML, describing the techniques and objectives, and with a general focus that goes beyond Machine Learning (ML), but instead embraces the wider AI field. The present scooping review spans on publications reported in Pubmed from 2003 to 2023, and resulting by searching “chronic myeloid leukemia” and “artificial intelligence”. The time frame reflects the real literature production and was not restricted. We also take the opportunity for discussing the main pitfalls and key points to which AI must respond, especially considering the critical role of the ‘human’ factor, which remains key in this domain.

## 1. Introduction

Artificial Intelligence (AI), is a set of methods and techniques expected to give a relevant contribution in many fields of the clinical domain. Querying Pubmed, for example, with “Artificial Intelligence”, from 1022 items in 2017 in only five years the number of items grew to 17,637. This is due to the variety of AI techniques and to their flexibility, supporting AI exploitation for a wide range of purposes, for example for building Computer-Assisted Image analysis systems or to assist/replace statistics in data analysis, for building accurate predictive models.

For instance, the United States food and drug administration (US-FDA) has approved a number of AI-based solutions form medical use since 2017 [1]. The introduction of digital pathology has brought many opportunities to the field of pathology, such as telemedicine. Recently, the use of digital pathology has fostered the use of Machine Learning (ML), one of the most promising subfields of AI, in the automation of pathological diagnosis. The challenges facing the use of AI in pathology are many, including digitalizing slides, labeling in case of supervised learning, initial and maintenance costs, advanced equipment, technical expertise, and ethical considerations. However, the potential opportunities for implementing AI and ML tools in pathology are numerous. For example: proposing new image classification hierarchies based on deep or handcrafted features found significant in the images; extracting features without supervision (i.e., labeled data); reducing the workload of pathologists on laborious and mundane tasks; becoming a low-cost (and hopefully highly skilled) second opinion provider, etc. [2].

The use and application of artificial intelligence methods for the diagnosis of common types and subtypes of leukemia has been explored over the last decades [3] and recorded significant successes, as reviewed by Salah et al. [4]. One of the most comprehensive efforts was accomplished in 2013 by using genome-wide expression profiling in the diagnosis and subclassification of different types of leukemias. Following this line of research, ML methods have then been proven useful for integrating large-scale-omics data from oncologic and oncohematologic patients, and for analyzing gene expression profiles in response to different drugs [5].

In Medicine, AI has been used for a range of different applications; these range from exact survival prediction in Chronic Myeloid Leukemia (CML) to individualized therapy recommendations, even though CML is generally seen as a clinical success in term of normal survival with therapy compared to other types of malignancy. [6].

In order to clarify the nomenclature concerning the present used acronyms, we summarized some of the main terms in Table 1.

## 2. Chronic Myeloid Leukemia

Chronic myeloid leukemia (CML) is a clonal myeloproliferative disease that affects one in ten thousand people each year. It is characterised by a reciprocal translocation of chromosomes 9 and 22 [t(9;22)]. It causes Philadelphia chromosome-positive (Ph+) and the formation of new fusion genes encoding for the chimeric BCR::ABL1 tyrosine-protein kinase (BCR::ABL1) which may present different isoforms: e.g., p190, p230, and p210 oncoproteins [14]. This latter is the most common in CML patients. BCR::ABL1 has constitutively high tyrosine kinase activity and is considered the hallmark of CML Ph+ because it is involved in disease’s pathogenesis and progression [15]. CML is divided into three stages: the chronic phase (CP), the accelerated phase (AP), and the blastic phase (BP). Indeed, BCR::ABL1 reduction or ablation is required to avoid disease progression to advanced BP [16]. Currently, the only treatment that can successfully control the progression of CML to the BP is tyrosine kinase inhibitors (TKIs) targeting the BCR-ABL1 p210 protein, inducing a significant reduction in the expression of the *BCR::ABL1* transcript, referred to as the major or deep molecular response (DMR), in 80–90% of patients [17]. The remaining patients are refractory or resistant to TKIs, and they pose a significant challenge in CML [18]. The *BCR::ABL1* transcript detection and quantification in peripheral blood (PB) cells using reverse transcription-quantitative PCR (RT-qPCR) and normalization to a housekeeping gene is recognized as the international standardized method for determining minimal residual disease (MRD) and plays an important role in the management of CML patients [19,20]. As aforementioned, the MRD level is distinguished as a major molecular response (MMR or MR3.0), with *BCR::ABL1* ≤0.1% and ABL1 >10.000 copies; or a DMR (MR4.0, MR4.5, MR5.0) if *BCR::ABL1* ≤0.01%, or ABL >10.000 copies when *BCR::ABL1* is undetectable [11]. The achievement of the DMR is associated with survival and the opportunity for treatment-free remission (TFR), that means the maintenance of molecular response without therapy [21,22,23,24]. Nevertheless, numerous studies have demonstrated the persistence of CML leukemic cells in the bone marrow (BM) niche following treatment, even in patients with undetectable levels of the *BCR::ABL1* transcript by RT-qPCR. The persistence of these cells together with leukemic stem cells in patients with CML has also been confirmed in clinical practice, where molecular relapse is experienced in ~50% of patients undergoing TKI discontinuation programs [18,25,26,27,28,29]. The introduction of digital PCR (dPCR) for MRD monitoring for research application only has improved the capability to select patients eligible for TKI discontinuation/de-escalation programs [30,31,32,33,34,35], even if dPCR standardization is still ongoing [36]. Moreover, dPCR has been still used for the detection of point mutations associated with TKI-resistance [37]. Despite that, additional variables need to be considered and a multiparametric evaluation accounting different parameters and a large amount of data would be the key.

In recent years, an increased interest in the application of AI in the CML settings has been observed, as proved by the growing number of publications on Pubmed (Figure 1).

In the following sections, we are presenting and commenting on different applications of AI in the context of the diagnosis and prognosis, considering different data set as input (imaging data vs. biochemical, biomolecular and clinical data), and treatment of CML.

## 3. AI for the Improvement of the Diagnosis and Prognosis of Adult CML Starting from Imaging Data

The present CML diagnostic workflow is based on a bone marrow (BM) aspirate for the evaluation of morphology, together with a core biopsy that may support the evaluation of fibrosis and additional nests of blasts not evident in the aspirate. Cytogenetics on bone marrow cells should be performed for the identification of the Ph chromosome. Then, a qualitative reverse transcriptase PCR on peripheral blood cells will identify the type of *BCR::ABL1* transcripts. If a molecular assay demonstrates the presence of *BCR::ABL1* fusion gene, but the Ph chromosome cannot be identified by cytogenetics, a FISH test is required. The diagnostic workflow is completed by a physical examination of spleen and liver, a standard biochemical profile, and an electrocardiogram [38]. So, the diagnosis of CML is mainly based on the identification of the Ph chromosome and the morphologic analysis. In parallel, a precise prognostication of the patients is essential for the therapy decision and monitoring strategies management. In fact, the prognosis focuses both on therapy response and on disease evolution to AP and BC. Over the years, three different scores have been developed: Sokal score, Euro score, EUTOS score, and ELTS score. Sokal and Euro scores were developed before the advent of TKIs. Sokal score considers age, spleen size, platelet and blasts count as variables, while Euro score also includes basophil and eosinophil counts. EUTOS score, developed after the introduction of TKIs therapy, considers only basophils counts and the spleen size [39]. Finally, EUTOS long-term survival (ELTS) that classifies patients in 3 categories, based on their probability to die because of CML [40]. The prognostication of CML is far from precise, especially after the introduction of the second generation TKIs and the TFR considered as one of the main goals. AI strategies have demonstrated to strongly improve the diagnosis and prognostication of different solid tumors and to support decision makers [41,42,43]; thanks to these successes, we are close to the turning point of AI introduction in clinical practice [44,45,46]. This does not appear to be the case for CML.

Despite a very recently application of ML to RNA sequencing analysis on 2606 samples, that is aiming at improving the application of next generation sequencing and AI for diagnosis and classification of different hematological malignancies [47], the application of AI strategies in the CML scenario has been actually applied mainly in the evaluation of the morphology of malignant cells in the BM aspirate. The first pioneering experience was reported 20 years ago by Swolin and colleagues. The Swedish group presented the DiffMasterTM Octavia system. It was composed by a microscope, a camera, an automated stage holder, an electronic hardware for motor and light control, and a software for automatic cell location and image processing. The software was developed for pre-classification of blood cells using artificial neural networks. The agreement between the test and the manual microscopy was 91%, whether the sample was abnormal or normal, while the sensitivity to identify blast cells was slightly higher with the DiffMaster than manual microscopy. Swolin and colleagues firstly demonstrated that a decision support system, together with a qualified morphologist, could generate leukocyte differential count reports of high quality and improved the diagnosis of hematologic malignancies, including CML [48].

This innovative approach opened the door to the application of AI to the diagnostic workflow of CML, even if 15 years were needed to obtain additional results. In fact, AI has been progressively investigated and applied with a special focus on the improvement and support of morphological analysis, and in parallel explored also in patients affected by lymphoid malignancies, as reported by a revision of literature in 201; the authors reported different results among the application of image processing and machine learning in the morphological analysis of blood samples [49]. Only one year later, Ahmed and colleagues included myeloid malignancies and CML in a study aiming at applying Convolutional Neural Network (CNN) to the identification of different leukemia subtypes. The authors used the data present in: ALL-IDB and The American Society of Hematology (ASH) Image Bank, 2 public repository databases, to test CNN architecture of deep learning for the identification of four subtypes of leukemia (Acute Lymphoblastic Leukemia (ALL), AML (Acute Myeloid Leukemia (AML), Chronic Lymphocytic Leukemia (CLL), and CML). The study demonstrated that CNN model presented 88.25% of accuracy in leukemia versus healthy analysis and 81.74% in multi-class classification of all leukemia subtypes. Moreover, the considered CNN model had a better performance than other machine learning algorithms well-known at the moment of the study [50]. The same dataset (ALL-IDB and ASH) has been used in another very interesting study in 2020. The Authors tested an Internet of Medical Things- (IoMT-) based framework to improve the identification of the 4 main subtypes of leukemia: acute and chronic myeloid leukemias, and acute and chronic lymphoid leukemias. Dense CNN (DenseNet-121) and Residual CNN (ResNet-34) were used and demonstrated 99.56% and 99.91% of accuracy, respectively. The proposed IoMT system, with the help of cloud computing, linked clinical gadgets to network resources providing a quick and safe leukemia diagnosis [51]. Working on a huge variety of dataset and published data, a recent publication reports a new algorithm able to identify the same 4 leukemia subtypes (AML, ALL, CLL and CML) by considering 25 features, confirming what previously observed [52]. These are very effective examples of AI applications allowing real-time coordination for leukemia diagnosis and treatment, which may save both efforts, costs and time of patients and healthcare professionals.

Efficiency has become of crucial importance after the pandemic scenario of COVID-19. In fact, in the last 3 years additional results have highlighted the potential of different and even more innovative strategies of ML as effective support to CML diagnosis. This confirms that it is a very hot topic. In example, another CNN application has been reported in 2020 for the diagnosis of AML, ALL and CML. The study included 104 BM smears, among which 18 resulted of CML cases. The authors applied 3 different CNN frameworks and improved the prediction accuracy of the model by transfer learning. The prediction accuracy of the application of CNN resulted 95% in the CML subset, confirming that the leukemic cell morphology classification and diagnosis by CNN combined with transfer learning is feasible. In addition, this method has been described as more rapid, accurate, and objective than conventional manual microscopy [53]. Another research group recently presented a conditional generative adversarial network (cGAN)-based model for the morphological analysis of bone marrow biopsy. In particular, the AI model was developed to segment megakaryocytes from myeloid cells and their statistical characteristics were extracted and compared between CML patients and controls. The presented cGAN was compared with 7 other deep learning-based models and resulted with a better segmentation performance. At the clinical validation phase, images from 58 CML cases and 31 healthy subjects were tested and confirmed the high accuracy of the cGAN-based model [54]. In a different work, Dese and colleagues reported similar results. They tested a ML-based automated optical image processing system on 250 blood smears. The samples corresponding to different acute and chronic leukemia subtypes and the ML-based system presented an accuracy, sensitivity, specificity of 97.69%, 97.86% and 100%, respectively for the test datasets, and 97.5%, 98.55% and 100%, respectively for the validation datasets in classifying leukemia types [55].

These are very impressive results, particularly when considering that the targeted study periods spanned October 1999 to April 2020 and February 2014 to December 2020. In fact, the quality and accuracy of the blood cell count technologies have been continuously improved and the technologies themselves have changed over the time. Considering the desirable future sensitive, accurate and precise techniques for blood analysis, the application of AI for an early diagnosis of CML as well as other hematological malignancies seems imminent. These assistant predictive tools in the decision support system could reduce the frequency of extra and irrational diagnostic tests which result in time-consuming extra burden on patients and laboratory staff.

## 4. AI for the Improvement of the Diagnosis and Prognosis of Adult CML Starting from Biochemical, Biomolecular, and Clinical Data

AI applied to diagnosis and prognosis of CML has been tested also considering data different from imaging data. In particular, different studies have been focusing on biochemical, biomolecular and clinical data as dataset input for a large variety of AI tools. The first attempt was reported by Dey and colleagues in 2011, more than 10 years ago. The authors described the application of a commercially available artificial neural network (ANN) software program to 40 CML cases who presented either AP or BC. Patients were divided into 2 study groups on the basis of the time of disease evolution: within 18 or 30 months from diagnosis. Considering clinical, hematologic, and morphometric data, the ANN software successfully classified the patients in the two groups. It meant that a commercial software, not specifically developed for CML data interpretation, was able to predict an early or late disease progression [56]. This early success of an AI program in classifying prognostically good and bad cases of CML supported subsequent studies. AI tools have been tested for the prediction of both relapse or progression as well as response to TKI treatment. In fact, in 2013 Ni and Colleagues presented the results of a support vector machine (SVM) strategy for the improvement of flow cytometry in the identification of malignant neutrophils in adult CML cases. SVM are supervised algorithms used to learn from samples (training group) and classify unknown objects (test groups). These can be used to simultaneously analyze multiple parameters, such as flow cytometry data, and to classify two sets of data in an n-dimensional space. Flow cytometry is quite limited in CML since pathologic and normal neutrophils present similar antigens. The authors used 18 adult CML cases as training group, while 67 newly diagnosed CML patients served as test group and LIBLINEAR was applied as SVM. The built predictive model differentiated between pathologic and normal neutrophils with specificity and sensitivity >95% [57]. Despite the impressive results, flow cytometry has not yet found a major role in the CML diagnostic workflow. Another very interesting study linked quantitative time-course information to disease outcomes aiming at improving the predictions for patient-specific treatment responses. The authors provide an in silico simulation of 5000 different patients’ response kinetics that was considered an artificial Acute and Chronic Myeloid Leukemia patient cohort. Then, they tested three different computational methods: mechanistic models, generalized linear models, and deep neural networks. In the subset of CML simulated patients, the time of therapy cessation was sampled based on kernel density estimates from the cessation time of the given patients. The authors used this information in de novo forward simulations. They generated artificial time-courses of varying duration until TKIs therapy stopped plus 10 years thereafter. CML molecular relapse was defined as MMR (or MR3.0). The prediction accuracy declined for all approaches when the data quality decreases, indicating that the lower the number of measurements, the lower the capability to predict patients’ relapse. Then, models were also forced by considering 1 year of TKI dose reduction, as it is commonly provided in clinical practice. Probing the system’s response to this perturbation gave additional information about control mechanisms that could not be obtained from ongoing monotherapy. The reported evidence confirmed that different computational approaches are in principle suited to support relapse prediction and, in particular, neural networks present good performance for the disease relapse prediction if frequent measurements are available, as in the CML case [58]. Similarly, Shanbehzadeh et al. used the data of 837 CML patients to test 8 ML tools, including eXtreme gradient boosting, multilayer perceptron, pattern recognition network, k-nearest neighborhood, probabilistic neural network, support vector machine (kernel = linear), support vector machine (kernel = RBF), and J-48. Features were divided into 2 data sets: “full features” and “selected features”. The latter consists of important features selected by minimal redundancy maximal relevance feature selection. Among the 8 considered algorithms, support vector machine (kernel = RBF) had the best performance on selected features with 85.7%, 85%, and 86% of accuracy, specificity, and sensitivity, respectively. The performance was reduced at accuracy of 69.7%, specificity of 69.1%, and sensitivity of 71.3% when full features were considered [59]. The reported studies support the hypothesis of an AI-driven prognosis for the prescription of personalized medicine for CML patients. In the same period, AI was also exploited in predicting future diagnosis of CML. A total of 1623 patients with BCR-ABL1 analysis to diagnose CML, and at least 6 consecutive years of varying blood cell counts before the BCR-ABL1 test, were retrospectively included either in the train/validation or test group. The variable selection was based on 2 different models to improve the strength of the study: a model based on Decision Trees (DT) based algorithm (XGBoost), and a modeling technique based on Logistic Regression (LR) based algorithm (LASSO). The minimum basophil percentage resulted predictive up to 1 year prior the BCR-ABL1 test and it was the only variable considered in the model of prediction at 1 year and 6 months before the CML diagnostic test. The optimal performance of the models was obtained at the moment of BCR-ABL1 test. Nevertheless, models trained on data gathered surprisingly at either 6 months or 1 year before the CML diagnostic test performed similarly. The performance of the model declined progressively when considering data acquired more than 2 years before BCR-ABL1 test [60]. Haider and colleagues reported comparable observations 1 year later [61]. In this case, the model used the potential morphological and immature fraction-related parameters generated during blood cell count, through AI/ML predictive modeling for early differentiation of the 4 main subtypes of leukemia considered also by Bibi et al. [51]. Haider and colleagues collected the data of 1577 patients diagnosed with hematological malignancies and developed an artificial neural network (ANN) predictive model. The Radial Basis Function Network (RBFN) tool was chosen and applied to the entire cohort. The ANN tool presented the highest correct classification performance really in CML cases: 90.1% and 97.5% in the training and testing set, respectively. Conversely, the lowest correct classification performance was observed in acute promyelocytic leukemia cases [61].

The therapy itself may be optimized and tailored via AI techniques, for instance to estimate the response or resistance to drugs as well as to select the best treatment strategy. Patients predicted not to achieve MR3.0 within 24 months to first line imatinib treatment may be better treated with second generation TKIs, such as nilotinib or dasatinib. Banjar and colleagues developed predictive models based on adult CML patients treated with imatinib and that either achieved or not MR3.0. The authors divided the cohort of patients into 2 subgroups which served as datasets for the ML models based on classification and regression trees (CART). Clinical, molecular, and peripheral blood factors were considered as well as predictive assays. Six different models were developed during the study. All the models presented a positive predictive value higher than the other considered conventional scores (73–96% vs. 67%). Taking into account a range of parameters, model D was the best one, and it was then externally validated. Although the common prognostic risk scores (Sokal, Euro, and Eutos) presented high accuracy, highest specificity (35%) was found in the ML-based model. It confirmed that the developed ML-based model accurately predicted the negative group (patients who will not achieve MR3.0 at 2 years) [62]. The use of AI as an additional tool to provide improved and personalized treatment plans to patients was investigated also by Sasaki and colleagues from the University of Tokyo and Houston. The research group presented the LEukemia Artificial intelligence Program (LEAP), an extreme gradient boosting decision tree-based method for the optimization of the TKIs treatment in adult CML patients in CP. The study enrolled 504 patients in the training/validation cohort and 126 in the test cohort and considered 101 variables collected at diagnosis. The considered therapy options were: imatinib (alone or with pegylated interferon), dasatinib, nilotinib, and ponatinib. Again, the study demonstrated the capability of AI in supporting clinicians’ activity by suggesting treatment which results associated with a better survival probability when compared to treatment not suggested by the LEAP-model [63].

All the presented results support the idea that ML models, supervised by human experts, and further AI-based innovations could effectively drive the improvement of patient outcomes and reduce the time of development of new prognostication strategies. The next few years will probably present a wider application of AI tools and the development of user-friendly ones, in order to facilitate their use by clinicians.

## 5. AI for a Personalized Management of the Therapy in Adult CML

The treatment strategy selection in adult CML patients may itself be an object of implementation and optimization by AI-models, as emerged by some of the studies presented in the previous section [62,63]. The prognostication of adult CML patients is strictly linked with the therapeutic strategies. These, mainly based on TKIs selectively targeting BCR-ABL1 protein, influence the different kinetics of leukemic burden reduction, the quality of life and the possibility to achieve the treatment free remission. Padhi and Kothari presented in 2007 the first attempt of using mathematical models and bioinformatic for the optimization of TKI therapy in CML patients. The strategy was based on the combination of optimal dynamic inversion and model-following neuro-adaptive control design, and aimed at designing an automatic drug administration to maximize efficacy of CML therapy. The method has two components: a nominal controller and a neuro-adaptive controller. The first is based on optimal dynamic inversion. The authors obtained encouraging results by a simulation based on artificial data for nominal patients, but it is unlikely that patients in real life will have the same parameters as used in the simulation. Hence, they perturbed all the parameters within the function in the model and randomly selected numerical values. The AI-based CML treatment strategy was very effective to treat the realistic model patients the presented techniques was so general to be applicable to any other similar nonlinear control design problem [64]. Another interesting example of ML applied to the optimization of therapy efficacy was presented more than 10 years later by Borisov and colleagues. They introduced an innovative method able to predict clinical efficiency of anti-leukemic drugs in patients by transferring characteristics obtained from the gene-expression analysis from different cell lines. An outstanding pioneering example of personalized medicine. The authors considered target therapies and tested 3 data sets: 1 of CML data and 2 of solid tumors data. Among CML, a total of 28 samples treated with imatinib were included: 16 responders, and 12 non-responders. The study was based on three different ML methods (SVM, binary trees, and random forests (RF)) to build predictor-classifiers. RF resulted not suitable as data transfer technique in the considered dataset, while the optimal data transfer parameters of SVM and binary trees allowed a correct separation of clinically responders from non-responders [65]. Similarly, a ML-method was applied to perform a sub-analysis in the context of the ENEST clinical trial. In this case, an ML-driven strategy identified differentially expressed key microRNAs as predictive biomarkers of nilotinib response. Data from 58 patients before and after nilotinib treatment were considered. The study was based on a survival statistical analysis combined with RF and bayesian ML algorithms and confirmed the capability of AI in supporting the optimization of CML treatment by dissetting responders to non-responders [66].

At the same time, some studies focused on the other side of the coin: drug resistance. Liu et al. first reported the combination of a single cell mass spectrometry analysis focused on cell metabolism with different ML-based strategies: RF, ANN and penalized LR. The predictive accuracy of each ML model on single cell metabolomic datasets was analyzed. ANN was superior to the other tested models. One of the main limits of this study is the origin of the input data: all the tests were conducted on cell lines data [67]. Also, Melge and colleagues used CML cell lines to perform an in vitro validation of 2 different machine learning supervised models. In this case, the authors applied AI to design a new drug combining 2 different molecules, one of which targeting BCR-ABL1 (namely ponatinib). Different chemical compounds were developed by the ML-supervised models, but the most promising one demonstrated to inhibit the growth of the cell lines, both in case of TKI-sensitive and TKI-resistant cells [68]. The presented approach has to be considered an interesting example of how AI, and ML in particular, can be incorporated as part of the CML investigation. Moreover, AI may support overcoming some of the challenges of CML in 2023: the arbor of TKI resistance. Another group faced the problem, too. They acknowledged that few efforts have been put into creating instruments to precisely identify ABL1 resistance mutations or elucidating their molecular underpinnings. As a result of the discovery of mutations that reduce the affinity of type I and type II inhibitors, SUSPECT-ABL, a revolutionary web-based diagnostic tool for anticipating resistance dynamics was created. The variations in ligand affinities caused by resistance mutations in ABL1 were successfully discovered and calculated. The method has identified potentially new resistance mutations by an in silico saturation mutagenesis, providing prospects for in vivo experimental confirmation. The proposed approach appears to be a crucial instrument for enhancing precision medicine initiatives as well as for accelerating the creation of inhibitors of the next generation that are less likely to acquire resistance. The authors have made the tool freely available on the web, giving the possibility to test it to the scientific community [69]. Similarly, Jie Su et al. [70] applied AI to develop new therapeutic molecules against T315I resistance-related mutation and confirmed all the result via in vitro cultures. In particular, they observed cycle arrest, autophagy and apoptosis, as well as inhibition of BCR-ABL1 phosphorylation.

Even if additional studies must be conducted to confirm the presented results on the more complex specimens sampled from CML patients, the response and resistance are not the only parameters to be considered for CML therapy. The side effects due to TKI therapy are very frequent and sometimes drive the switch to other TKIs as well as the discontinuation of the treatment. In 2022, an indirect application of AI for the identification of TKI-associated side effects was presented. A novel text mining tool was used to predict adverse events, under-reported or preclinical ones included, using 2575 clinical abstracts about CML-TKI as input data. The authors put forward a new cross-domain text mining technique that forecasts new relationships using a knowledge graph, link prediction, and hub node network analysis. Methods included two significant analyses: (a) Novel cross-domain text mining over all 30+ million biomedical papers in PubMed to detect and rate less well-known potential adverse events (AEs) connected to tyrosine kinase. (b) Bag-of-words cluster analysis to connect adverse events to specific TKI medication classes. Known and new TKI AEs were predicted and ranked using cross-domain text mining. Using the feature importance of unsupervised rank aggregation, three physiology-based levels were created. The findings point to regular surveillance for tier 1, rarely surveillance for tier 2, and symptom-based surveillance for tier 3 as proactive TKI patient AE surveillance levels. Cross-domain text mining via natural language processing (NLP) and ML enable expanded types of exploratory analyses that would not be otherwise possible [71].

Considering the success of TKI selectively targeting BCR-ABL1 protein, it is not surprising that the vast majority of AI applications focuses on those molecules’ effects and interactions. Despite that, an interesting study proposed the application of AI for the identification of the pharmacological pathways of clumps of dry powder, obtained by machining the leaves or stems, namely Qingdai. Qingdai is traditionally used for CML treatment following traditional Chinese medicine. Li and colleagues applied network pharmacology approaches and identified the candidate Qingdai targets according to their network topological importance. Three types of visual networks were built: the compound-target network, the target-pathway network, and the target-target network. The results, in terms of components and corresponding candidate major targets, were further validated by a molecular docking simulation. Seven components in Qingdai were selected and 32 candidate Qingdai targets identified. All of them play important roles in the progression of CML [72]. Despite further clinical application, assessments and experimental validations for these predicted results are required, this explorative study supports the use of AI-based tools to identify and predict new molecular interactions improving the CML target therapy and the combined-therapy strategies. In fact, Naveed and colleagues recently published an elegant AI-based study on the natural vitamin E molecule gamma-tocotrienol as a BCR-ABL1 inhibitor. Different tools were used: AlphaFold (for the prediction of 3D structures of proteins using a deep learning model), DeepSite (a DNN modulator for protein binding pockets analysis), Small Molecule Suite (a tool for chemical compound selection), WADDAICA (a web server for the alteration or generation of different compounds), and ProTox-II (a virtual tool to predict toxicity of small molecules). The authors created 3 potent de novo therapeutic molecules for the BCR-ABL1 chimeric protein using a deep learning artificial intelligence (AI) drug design technique. When the AIGT was docked with BCR-ABL1, it showed a binding affinity of −7.486 kcal/mol, suggesting that it could be a viable pharmacological option and that it may have hepatoprotective properties [73].

The timeline of the cited studies, reported also in Table 2, is graphically represented in Figure 2.

Considering the presented results, it is indisputable that AI, and ML in particular, represents an innovative way to support CML treatment and to examine in depth the pitfalls of the current therapeutic strategies, as well as the identification of new mechanisms and targets. Moreover, AI tools could avoid the application of high-throughput screening experiments, which often are expensive and time-consuming, driving the scientists to an efficient validation of the insights.

The clinical trials commented on the previous sections has been summarized in Table 2.

## 6. AI for Knowledge Representation

Even if many of the most well-known AI applications are based on ML techniques, there are other AI disciplines that are growing in importance and interest. Amongst the most representative, for example, are formal ontologies. Ontologies are of particular interest in the field of automated reasoning, where an agent is aimed to be able to perform logical inferences on the basis of some known facts of the word (e.g., “Sam Smith has been admitted yesterday”) and some general rules. From these building blocks, the use of a dedicated ontology encoding some knowledge of the medical domain (e.g., “every admitted person is a patient”), can therefore infer that Sam Smith is a patient without an explicit human declaration. The example is trivial, but since every deduced fact can, in turn, generate other cascading deductions, the chain of inferences can progress to highlight a large quantity of logically deducible truths, well beyond the capacity of human inference. For example, from a few basic, not specifically targeted pieces of information, the system could deduce a dangerous drug interaction for a specific patient. Even though we are far from having systems of this kind in everyday clinical practice, some elements, at the ontological level, are already available. Querying the website bioportal (https://bioportal.bioontology.org/, accessed on 12 January 2024, one of the most common repository of clinical ontologies) with the string “Chronic Myeloid Leukemia” retrieves 31 ontologies, ranging from generalist ontologies such as SNOMED to ontologies designed for more specific tasks, such as the “Biological Pathway Taxonomy” (https://bioportal.bioontology.org/ontologies/BPT, accessed on 12 January 2024) or the “Ontology of Drug Adverse Events” (https://bioportal.bioontology.org/ontologies/ODAE, accessed on 12 January 2024). These ontologies, often represented by devoted languages such as OWL (https://www.w3.org/OWL/, accessed on 12 January 2024), can be parsed by the so called Semantic Reasoner like JENA [74], PROVA [75], FLORA-2 [76]. While this class of approaches is not yet as well received as ML-based ones, it holds great promise in terms of benefits: ontologies can encode vast amounts of knowledge, and can be validated by human experts—supporting trustworthiness and explainability, and can be extended if additional information are made available. Further, they can be used in combination with ML approaches, to generate synergies.

## 7. Discussion and Conclusions

On the basis of the evidence of the analyzed work, we can confirm diffused expectations about the emerging importance of AI in supporting clinicians at many levels, in coping with Chronic Myeloid Leukemia, as reported also by Elsabagh and colleagues [77].

Even if the use of Artificial Intelligence in this application field is still limited, it is soon expected to deliver benefits in a number of activities, such as in supporting the diagnostic process or assisting the definition of the clinical strategies. In supporting the diagnostic process, for example, the field of automatic contouring is encouraging because the quality and accuracy of the blood cell count technologies have been continuously improved, and this can lead to more time/money efficient diagnostic processes [49]. Similarly, image analysis can be used for diagnostic purposes with significant performance [50,55]. Looking at the future, we can see promising avenues for future improvements: the ever-growing number of studies, in Computer Science, about ANNs based-algorithms (boosted by the adoption of the GPUs designed for gaming), is delivering impressive results in pivotal (highly funded) clinical areas [78,79] and this is likely to have repercussions in many other clinical fields.

The development of Decision Support Systems for prognosis is in the spotlight of a large number of research centers (often thanks to the growing availability of structured data). This is because AI is particularly effective in manipulating large numbers of covariates by overcoming (under specific circumstances, e.g., strong non-linearity) some limitations of classical statistical analysis tools. Nowadays, even if the results are not easily reproducible, some interesting insights can be found in [62,63]. In addition, the possibility to work with high-dimensional feature spaces fits well with the goal of Personalized Medicine, because it allows to take into account a very detailed description of the clinical case to evaluate.

However, it is crucial to understand the risks that can come from the use of AI techniques to prevent and mitigate them, ensuring a beneficial use of AI for everyone involved.

Currently, AI demonstrated the capability to provide remarkable performance in particular in Data Analysis: here it can generate accurate diagnostic/prognostic predictive models also exploiting partial/incomplete knowledge (e.g., with Bayesian Networks), or can explore correlations between huge sets of covariates beyond the limitations of the human cognitive patterns. Nevertheless, the gap between the classical, statistical analysis, and ML is technically relevant, and, from some points of view, it can be defined as *cultural*. This means that it is going to require a medium-long time to be properly “metabolized” by clinicians, who can dedicate only a limited part of their time in this training.

In the following, we provide some (challenging) aspects to consider:-A priori vs. a posteriori. While in statistics the experimental setting should be detailed a-priori (which test, which sample size, etc.). Often, this is because the common assumption of statistics (e.g., about the statistical distribution of the residuals) cannot be made at the beginning or, generally speaking, the a-priori approach is not efficient in order to get a reasonable model. On the other hand, ML has a more empirical approach, and it is more oriented to a-posteriori test the results. This requires a more critical approach in interpreting the results, due to the tangible risk of overfitting.-*Black Box*. In most cases, ML is not always able, by design, to show which features are relevant to make a prediction. This can happen for example with ANN or RF: in both cases, we might need to define some exotic new tools, such as the GINI mean index, to estimate the role of a covariate in the model. Even if those can give a sort of score, the mathematical meaning still remains quite obscure for the majority of the clinicians who are not trained in reading those indexes. Given the black box nature of most ML techniques, In recent years, an emerging field devoted to making AI more communicative, called Explainable AI, has been growing in importance. However, most of the effort is focused on post-hoc analysis of the results, by aiming at generating models that can mimic performance, rather than at explaining how the specific model actually internally works. On top of that, the rapid release of new techniques and approaches makes hard to achieve explainability and understanding of behaviors.-*Wild Wild West*. In looking at the experimental setting, we can observe a big heterogeneity of approaches, for example in the feature selection strategy or in the more critical validation/testing step. Unfortunately, there is a lack of standards in many steps of the so-called “computational pipeline” (the sequence of steps spreading from data collection to model performance assessment). Even if some standardization initiatives, such as the TRIPOD [80], or IBSI [45], are available, they are often not known or considered. This lack makes it difficult for a non-expert to understand the qualities of the experiments and data analyses, thus risking to create a misperception about the value of the results.-*Publish or Perish (PoP) culture meets market rules*. The pervasive bibliometric-based publish or perish politics has created a “social need” of quickly publishing high impact work, and the nature of AI perfectly fits with an exciting storytelling where science meets something which is in the order sci-fi. This, coupled with the fact that only a small part of the clinical-reviewers are concretely skilled in AI, lead to an inflation of reviews and a lack of original work in the field. As an example, querying Pubmed with the key “Radiomics” in 2019, retrieved 132 items classified as “Review” and “Systematic Review”. However, the same key retrieves only 36 items in the entire range 2015–2019 classified as “Clinical Trial” and “Randomized Controlled Trial”. This raw result is not definitive and surely needs to be better understood, but at a first glance, it might suggest the hypothesis that a lot of people, due to marriage between PoP culture and the easy enthusiasm, prefer to talk about AI instead of working with AI. This is not necessarily an issue: on the one hand, this risk to over-excite the community and creating a misperception and disaffection (see for example the AI Winter [81]), on the other hand, AI/ML is attracting funds and investments at a rate that was unthinkable only 15–20 years ago.-*Reproducibility of the results*. In particular, in Image Analysis, the problem of reproducibility is one of the most challenging issues. It can be difficult to estimate the role of the technology involved in the image acquisition and reconstruction, because the cognitive patterns of an AI agent can be obscure (especially if based on an ANN-family algorithm, as discussed above). Specific artifacts, which might depend on the set and version of implemented reconstruction algorithms, the quality of the hardware and some specific environmental features which can be not reproducible in other places may affect the results. This consideration can potentially be extended to any other technical device, which, in future versions, would be able to operate with more accuracy and, consequently, a different signal/noise ratio. This can also mean that the improvement in the signal quality has the potential to be seen as a noise from the point of view of a model trained with more obsolete technology. This would create an evident problem, due to the continuous technological improvement and would require the introduction of the concepts of “lifecycle” for each predictor.-*Ethics*. The evolution of AI poses critical ethical issues. While an extensive discussion is beyond the scope of this paper, it is worth mentioning this critical problem. One of the facets of the problem concerns the relation between the Decision Maker (DM) and the Decision Support System (DSS). Historically, the DM is the clinician while the software is a DSS, due to a mingling of culture and legal implications: the DM, after a consultation of the DSS, makes a choice according to his feeling (art) and knowledge (science). What would happen in a future scenario where the performance of a Black-Box DSS is empirically higher than the human? If the economic balance between risk and benefit (e.g., expected clinical performance, legal implication etc.) would depose in favor of trusting a decision proposed by an AI, how would the clinician’s profession change? Would the AI become a sort of colleague? How to mediate/interact with the patients in depicting this new environment? Do we need to train v2.0 physicians so that they can guide the evolution of AI and not instead be helplessly guided by it themselves? How can we change curricula, in this regard? Nowadays, in operating autonomously with human life, machines, robots, and AI suffer from the bias of a cultural taboo. However, sooner or later, performances might become comparable in making diagnoses or prognoses in some subdomains. Additionally, a robot never sleeps, is never tired, and is expected to be globally significantly cheaper. For this reason, we should be ready to judge without prejudice what could be better for us, the patients, and society [82].-*Humans are more than numbers.* Due to the need of features and data, AI can push the notion that patients are merely a set of numeric values to be assessed. However, patients have a human dimension that is challenging to express in numbers, but represents a significant source of inspiration for clinical decisions. It is not excluded that AI may, in some way, succeed in including some of these traits in its models in the coming years. However, at present, this sphere remains the prerogative of humans and is difficult to elicit because it is closely tied to personal experience, making sharing challenging. Large Language Models (LLM), can provide a promising approach [83] to help capturing some of the non-numeric traits of patients, but nowadays their application in medicine is still largely unexplored.-*Faster and faster.* Science is advancing rapidly, especially in the discovery, testing, and implementation of new biomarkers. This challenge is generally associated with data modeling, both in AI and statistics. However, AI agents are typically more autonomous in loading input data and providing a high-performing mathematical model. For example, they do not require a causal hypothesis a priori. Thanks to the technological nature of AI agents (the term ‘informatics’ derives from the French ‘*information automatique*’), they are expected to be easily integrated into a Data Treatment Ecosystem where data can be extracted from hospital Electronic Healthcare Records, and models can be updated with new evidence and new biomarkers [84].-*Cultural Taboo.* Nowadays, in operating autonomously with human life, machines, robots, and AI suffer from the bias of a cultural taboo. This for a good reason: they are not ready. However, sooner or later, performances might be comparable, in making diagnosis or prognosis, and we need to be ready for that [82]. A robot never sleeps, it is never tired, and it is expected to be, globally, significant cheaper.-*Bias.* As AI systems need to be trained or optimized on large amount of data, the resulting systems tend to be biased towards the population from which data has been collected. Currently, this is mostly from rich Western countries that have funds and infrastructure to support the deployment of data pipelines and data sharing environments. The clear implication of that is the potential for suboptimal treatment of underrepresented populations, and a widening gap between technologically advanced countries and the rest of the world.

These challenging aspects, together with the advantages, are reported in Table 3.

The listed points only represent what we believe are the most interesting and clinically engaging open issues and factors to be considered for the development of AI in medicine, with particular focus on CML. In addition to the previous mentioned points, we could list for example: internal technical issues (e.g.: computational issues, algorithms limitations), institutional issues (the need of high data quality, and robust software infrastructure), semantic issues (data harmonization on common data models or ontologies) or legal issues (patient’s privacy, data protection). It is evident that these themes, on the one hand, require specific and in-depth expertise, and on the other hand, the ability to work as a team. In the years to come, it is possible that this will transform the ways of working, interacting, and dialoguing, perhaps even creating new professional figures, in order to mature the potential of AI and transform it from the land of promises into the promised land.

## Figures and Tables

**Figure 1 cancers-16-00848-f001:**
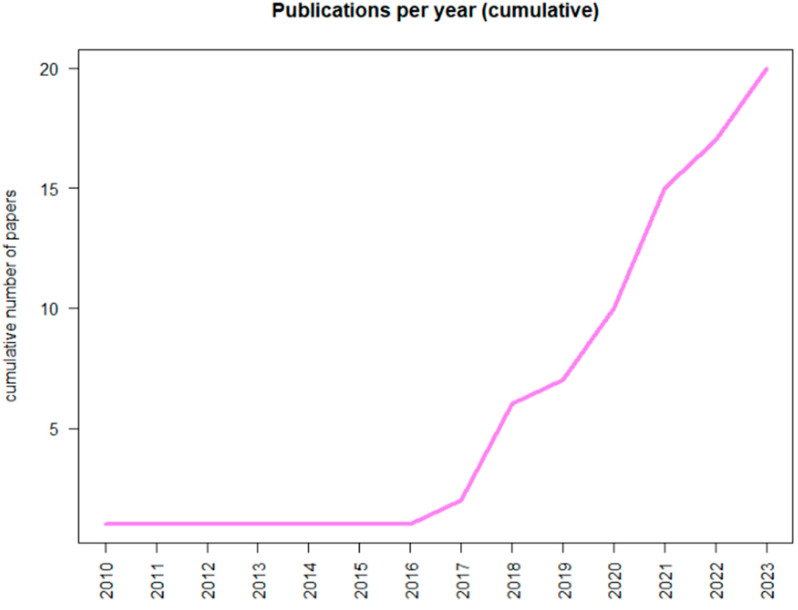
Graphical report about the number of publications concerning Chronic Myeloid Leukemia and Artificial Intelligence reported in PubMed.

**Figure 2 cancers-16-00848-f002:**
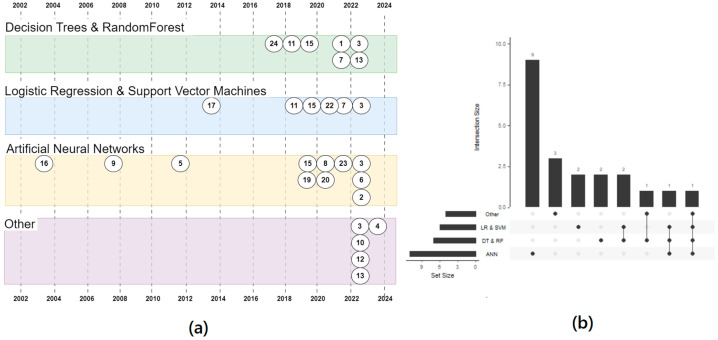
Timeline of clinical trial applying AI in the setting of CML. The trials are divided based on the AI strategies reported. (**a**) the timeline of publications by year and clustered by technique. The numbers refer to the IDs in the Table 2 (**b**) An upset plot showing how often the techniques were used. Mostly they have been used alone (the first 4 vertical bars: 9 + 3 + 2 + 2 = 16 times) and only 7 times more than one technique has been used in a paper. Only one paper tested all the four techniques (the last vertical bar, on the right). The horizontal bars show the usage of any cluster.

**Table 1 cancers-16-00848-t001:** The commonly used AI-related acronyms are reported and commented on.

Acronym	Meaning	Comment
AI	Artificial Intelligence	The discipline aimed at the development of intelligent autonomous systems.
ML	Machine Learning	Machine Learning is a subset of Artificial Intelligence (AI) concerned with creating systems that learn or improve performance based on the data they use.
SVM	Support Vector Machines	Is a Machine Learning algorithm for building classifiers and regressors based on the Support Vector Networks to find the hyperplane able to ensure the optimal margins in separating different regions of the space [7].
ANN	Artificial Neural Network	Generally speaking, the ancestor is a graph of perceptrons [8] where the connections are weighted (trained) by a backpropagation algorithm [9]. In the last years, many other kinds of ANN are emerged, overcoming the limits of the perceptrons of the original feed-forward connection.
RBFN	Radial Basis Function Network	An ANN that exploits radial basis functions as activation functions.
CNN, CANN	Convolutional Neural Network, Convolutional Artificial Neural Network	Is a recent evolution of the ANN, where a set of multi-layer convolution kernels are automatically trained on the input data [10] and, often, provide inputs for a feed-forward ANN. They are most commonly applied in image analysis/recognition/synthesis.
cGAN	conditional Generative adversarial network	cGAN [11] are a type of neural network that generates data conditioned on additional information, commonly used in image synthesis and modification.
IoMT	Internet of Medical Things	Define a complex network of hardware or software devices that connect and exchange data with other devices over the Internet or other communications networks in the Healthcare domain.
DT	Decision Trees	A family of ML classifiers/regressors topologically shaped as a tree (a connected acyclic undirected graph)
LR	Logistic Regression	A popular ML algorithm based on the application of a sigmoid function (e.g., a log-odds) on a linear regression model to easily obtain a classifier
CART	Classification and Regression Trees	see DT (Decision Trees)
BN	Bayesian network	A probabilistic model that represents a set of variables and their conditional dependencies via a directed acyclic graph. It allows for efficient inference of uncertain relationships between variables and usually assumes that the features are conditionally independent
NB	Naïve-Bayes network	A Naïve-Bayes network is one of the simplest kind of Bayesian Network (BN)
RF	Random Forest	A set of Decision Trees trained as regressors or classifiers. In a RF [12], the prediction of each DT is collected and the final prediction can be obtained by voting, weighted voting or more refined approaches.
kNN	k-nearest neighbors	A ML algorithm that predicts a class or a numeric outcome on the basis of the previous most similar k observations [13].
GA	Genetic Algorithm	A genetic algorithm is a search heuristic inspired by natural selection. It uses techniques such as mutation, crossover, and selection to evolve solutions to optimization problems

**Table 2 cancers-16-00848-t002:** Summary of the studies commented on the present Review. For ML techniques acronyms, refer to Table 1. Y = yes.

No	Study	Ref	Cases	Training/Testing Method	ML Techniques	Multicentric	Aims
1	Sasaki, 2021	[63]	630	cross-fold validation	DT	-	survival analysis
2	Zhang Z, 2022	[54]	89	cross-fold validation	ANN (cGAN)	-	automatic segmentation
3	Shanbehzadeh M, 2022	[59]	837	cross-fold validation	DT, kNN, ANN, SVM	-	survival analysis
4	Zhang H, 2023	[47]	46	cross-fold validation	BN	-	diagnostic classifier
5	Dey P, 2011	[56]	40	cross-fold validation + independent testing set	ANN	-	prognostic classifier
6	Haider RZ, 2022	[61]	213	training + testing set	ANN	-	diagnostic classifier
7	Hauser RG, 2 021	[60]	1623	cross-fold validation + independent testing set	DT, LR	Y	diagnostic classifier
8	Huang F, 2020	[53]	104	training + testing set	ANN		diagnostic classifier
9	Padhi R, 2007	[64]	-	-	ANN	-	PK/PD
10	Melge AR, 2022	[68]	-	-	2D-QSAR	-	molecular target
11	Borisov N, 2018	[65]	-	-	DT, SVM	-	drug efficacy
12	Mehra N, 2022	[71]	2575	-	clustering	-	text-mining
13	Yen R, 2022	[66]	58	training set	RF, BN, survival analysis techniques	-	prognostic classifier
14	Zhou Y,2021	[69]	-	-	-	-	-
15	Liu R,2019	[67]	-	training + testing set	RF, LR, ANN	-	prognostic classifier
16	Swolin B, 2003	[48]	322	pre-trained model	ANN	Y	cell counter
17	NI W,2013	[57]	65	training + testing set	SVM	-	diagnostic classifier
18	Rodellar J, 2018	[49]	-	-	-	-	image classifier
19	Ahmed N, 2019	[50]	903	cross-fold validation	ANN	-	image classifier
20	Bibi N,2020	[51]	1122	training set	ANN	Y	diagnostic classifier
22	Dese K, 2021	[55]	520	cross-fold validation + independent testing set	SVM	-	image classifier
23	Hoffmann H, 2021	[58]	275	training + testing set	GA/ANN	-	prognostic classifier
24	Banjar H, 2017	[62]	210	cross-fold validation + independent testing set	DT	-	prognostic classifier

**Table 3 cancers-16-00848-t003:** Summary of the advantages, disadvantages and limitation and expected future perspectives.

Advantages
New classification systems. Automatic feature extraction (from images, text, signals, etc.) Reduction of laborious and mundane task Low-cost and second opinion provider A different cognitive approach in mining knowledge from data
Disadvantages and Limitations
Black Box effect: how to elicit knowledge? From an a priori to an a posteriori validation mindset The Wild West of methods, in data analysis Publish or perish and low quality research Reproducibility of the results Ethics Humans are more than numbers The world of research is increasingly faster. Cultural Taboo Bias Additional technological challenges (algorithms, infrastructures, …)
Future Perspectives
A greater multidisciplinarity The emergence of new professional figures An improvement in data quality, harmonization, and their shareability to build models trained across multiple centers. Cost reduction

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
