# Peer review of "Artificial Intelligence-Based Management of Adult Chronic Myeloid Leukemia: Where Are We and Where Are We Going?"

_cancers, 2024, doi:10.3390/cancers16050848_

Round 1

Reviewer 1 Report

Comments and Suggestions for Authors

Bernardi et al summarized the current status of AI-based management in adult patients with CML. Overall, the manuscript is well-written to summarize the current status of the use of AI in cancer. This research will pave a way for the potential use of AI in cancer. I have a few minor comments.

Minor:

#1. Abstract: It is common misunderstanding to say CML is an easily managed malignancy. Survival has significantly improved in patients with CML since 2000. However, the patients may have chronic toxicities and long-duration therapy. I think the purpose is to mention expected normal survival with therapy compared to other types of malignancy. I would suggest to rephrase this part. 

 #2. I agree with the suggestions on the challenging points. I think it is worth mentioning a limitation of "datability". In clinical practice, healthcare providers pick important information from facial expression, voice tones, physical strength, patient's tendency of verbal symptom grading (underestimated, appropriate, overestimated), and reliability and consistency of patient symptoms. These parts are not converted into data for the production of AI models, at least currently. 

#3. Another challenging point is the progress in our understanding of disease biology. Science continues to progress and every year there are breakthroughs in biology. Essentially, there is no end. When potential prognostic variables were identified, the variables should be included in AI for further prognostification and treatment decisions. The development of AI in clinical care requires constant improvement though the improvement of accuracy could be minimal even with additional big data. 

Author Response

We thank the Reviewer for the very important and helpful suggestions which helped us in improving the quality of the manuscript.

Please, find below the point by point rebuttal letter.

Bernardi et al summarized the current status of AI-based management in adult patients with CML. Overall, the manuscript is well-written to summarize the current status of the use of AI in cancer. This research will pave a way for the potential use of AI in cancer. I have a few minor comments.

Minor:

#1. Abstract: It is common misunderstanding to say CML is an easily managed malignancy. Survival has significantly improved in patients with CML since 2000. However, the patients may have chronic toxicities and long-duration therapy. I think the purpose is to mention expected normal survival with therapy compared to other types of malignancy. I would suggest to rephrase this part.

Answer: Thank for this suggestion. The purpose was exactly what suggested by Reviewer. We rephrased the sentences in the abstract. Please, see line 15.

 #2. I agree with the suggestions on the challenging points. I think it is worth mentioning a limitation of "datability". In clinical practice, healthcare providers pick important information from facial expression, voice tones, physical strength, patient's tendency of verbal symptom grading (underestimated, appropriate, overestimated), and reliability and consistency of patient symptoms. These parts are not converted into data for the production of AI models, at least currently.

Answer: Thanks for the suggestion. Actually, yes: we agree this is one of the biggest limitations, currently. We have enriched the Discussion with some additional considerations in the point “Humans are more than numbers”. Please, see line 665

#3. Another challenging point is the progress in our understanding of disease biology. Science continues to progress and every year there are breakthroughs in biology. Essentially, there is no end. When potential prognostic variables were identified, the variables should be included in AI for further prognostification and treatment decisions. The development of AI in clinical care requires constant improvement though the improvement of accuracy could be minimal even with additional big data.

Answer: Thanks, we added a short comment about the (fortunately) never ending story of Science and the potential role of AI in Discussion (point: “faster and faster”). Please, see line 677

Reviewer 2 Report

Comments and Suggestions for Authors

1. AI-applicated CML Clinical Trials could be listed as a section. Results and FDA approvals would be helpful to understand the importance of AI to CML.

2. The advantage of AI over humans in diagnosis and treatment outcomes might be reviewed more extensively.

Author Response

We thank the Reviewer for the very important and helpful suggestions which helped us in improving the quality of the manuscript.

Please, find below the point by point rebuttal letter.

  1. AI-applicated CML Clinical Trials could be listed as a section. Results and FDA approvals would be helpful to understand the importance of AI to CML.

Answer: We really thank the Reviewer for this very interesting point, but we believe that a mature scouting, classification, and discussion of this would sincerely go beyond our aims. However, because your idea is brilliant, we have inserted in conclusion that “The rapid changes in AI require being more proactive in staying aware of ongoing projects. For example, a solid framework capable of scouting approved clinical trials by major institutions (e.g., FDA, EMA, etc.) and reporting on ongoing studies would be crucial.” Please, see Discussion section.

  1. The advantage of AI over humans in diagnosis and treatment outcomes might be reviewed more extensively.

Answer: This is a very important point. Unfortunately, even though the cited papers clearly exposed the performance of the AI/ML models, almost all omitted to mention the performance of humans in achieving that goal. For this reason, a quantitative comparison on the base of the reviewed papers is not possible. However, we think that a deeper discussion about expected pros and cons of Machine vs Humans in diagnosis/prognosis is helpful. We have stressed this point in our Discussion, also citing an additional paper. Please, see lines (659-664).

Reviewer 3 Report

Comments and Suggestions for Authors

Dear Authors

Below I present point by point comments and suggestions regarding your paper:

- English should be checked once again preferably by the native speaker since it requires significant corrections as numerous grammatical errors have been detected during revisions

- in the abstract it could be mentioned just to provide more detailed information, in which medical specialities AI is currently most widely used. Further in the abstract you should provide more information about the type of the manuscript and search strategy (you could indicate the time frames of your search of the articles included in the review, language of the reviewed articles, etc.). In this form the abstract is quite unacceptable since there is only a brief introduction, then aim and immediately the conclusions. It requires correction.

- Lines 41-42 about the opportunities of the implementation of AI in the ML - could you provide more detailed information about this for instance in the form of a table?

- Please unify the heading for your figures/tables - sometimes you write 'Fig.' sometimes 'Figure'. Please look at the author guidelines and adjust your paper to the recommendations of the Journal

- Table 2 - you should also provide the separate column for the references

- Line 584 - till the end. Please remove 'To briefly recap'. Instead write a solid paragraph with all the conclusions

- It would be beneficial to add a table summarizing all your findings regarding the application of AI in ML. It should contain advantages, disadvantages, limitations and future perspectives as well.

Regards

Comments on the Quality of English Language

- English should be checked once again preferably by the native speaker since it requires significant corrections as numerous grammatical errors have been detected during revisions

Author Response

We thank the Reviewer for the very important and helpful suggestions which helped us in improving the quality of the manuscript.

Please, find below the point by point rebuttal letter.

Dear Authors

Below I present point by point comments and suggestions regarding your paper:

- English should be checked once again preferably by the native speaker since it requires significant corrections as numerous grammatical errors have been detected during revisions

Answer: Thank you for this suggestion. The text has been edited for English languages and grammar.

- in the abstract it could be mentioned just to provide more detailed information, in which medical specialities AI is currently most widely used. Further in the abstract you should provide more information about the type of the manuscript and search strategy (you could indicate the time frames of your search of the articles included in the review, language of the reviewed articles, etc.). In this form the abstract is quite unacceptable since there is only a brief introduction, then aim and immediately the conclusions. It requires correction.

Answer: Thank you for this important suggestion. In order to improve the quality of the Abstract, the time frames and the key words used for the revision of literature are now reported in the Abstract. Please, see lines 25-28.

- Lines 41-42 about the opportunities of the implementation of AI in the ML - could you provide more detailed information about this for instance in the form of a table?

Answer: Thanks. The mentioned paper touched the ‘many opportunities’ from many perspectives, some of which strongly connected to the technology of AI (maybe a bit over the interests of the audience). We have extracted the most significant for a clinical audience and we have exposed them in a narrative form. Please, see line 52-57.

- Please unify the heading for your figures/tables - sometimes you write 'Fig.' sometimes 'Figure'. Please look at the author guidelines and adjust your paper to the recommendations of the Journal

Answer: We thank the Reviewer for this point. We correct Fig in Figure all over the text.

- Table 2 - you should also provide the separate column for the references

Answer: According with Reviewer’s suggestion, a new column reported the references was added to the Table 2

- Line 584 - till the end. Please remove 'To briefly recap'. Instead write a solid paragraph with all the conclusions

Answer: Thank you for this suggestion. We added a new paragraph reporting all the conclusions and our critical point of view concerning also additional technical issues and the multidisciplinary approach that AI requires. Please, see the Discussion section.

- It would be beneficial to add a table summarizing all your findings regarding the application of AI in ML. It should contain advantages, disadvantages, limitations and future perspectives as well.

Answer: Thank for this important suggestion. We added a table reporting advantages, disadvantages, limitations and future perspectives, as required. Please, see Table 3.

Regards

Reviewer 4 Report

Comments and Suggestions for Authors

This is an interesting paper in a field that needs more research.

Please engage a proficient english biologist/medical specialist to correct style and grammar.

For example:

 Line13. AI is taking its first steps, although the results obtained in the initial 13 experiments are interesting and promising both in terms of performance and the diversity of con-14 texts in which AI can be applied.  

Line 52. Among the various applications, AI and ML have been interestingly explored from 51 the personalized recommendation on therapy to a precise survival prediction also in 52 Chronic Myeloid Leukemia (CML), despite this disease is commonly considered a clinical 53 success [6].  

Could use punctuation/clarification.  CML is not a success as the sentence implies.

Line 63.  It causes Philadelphia chromosome-positive (Ph+) CML 63 and the formation of new fusion genes encoding for the chimeric BCR-ABL1 tyrosine- 64 protein kinase (BCR-ABL1) p210 oncoprotein.

This is oversimplified.  Avoid specifics since there are many different fusion proteins.

Line 122. Sokal score, Euro score, and EUTOS score.

What about the ELTS?

Line 505. he clinical case to evaluate. 504 However, the world is not black and white, and it is important, one step before a 505 possible revolution, to be able to identify the pitfalls near the gold nuggets.

Line 584. sneaky

Not standard scientific jargon.

This previous reference seems pertinent and not included:

Elhadary M, Elsabagh AA, Ferih K, Elsayed B, Elshoeibi AM, Kaddoura R, Akiki S, Ahmed K, Yassin M. Applications of Machine Learning in Chronic Myeloid Leukemia. Diagnostics (Basel). 2023 Apr 3;13(7):1330. doi: 10.3390/diagnostics13071330. PMID: 37046547; PMCID: PMC10093579.

Comments on the Quality of English Language

As above.

Author Response

We thank the Reviewer for the very important and helpful suggestions which helped us in improving the quality of the manuscript.

Please, find below the point by point rebuttal letter.

This is an interesting paper in a field that needs more research.

Please engage a proficient english biologist/medical specialist to correct style and grammar.

Answer: Thank you for this suggestion. The text has been edited for English languages and grammar.

For example:

Line13. AI is taking its first steps, although the results obtained in the initial experiments are interesting and promising both in terms of performance and the diversity of contexts in which AI can be applied. 

Answer: Thank you for this point. We rephrased the sentence. Please, see lines 15-16

Line 52. Among the various applications, AI and ML have been interestingly explored from 51 the personalized recommendation on therapy to a precise survival prediction also in 52 Chronic Myeloid Leukemia (CML), despite this disease is commonly considered a clinical 53 success [6]. 

Answer: Thank for this suggestion. The sentence has been rephrased. Please, see lines 68-72

Could use punctuation/clarification.  CML is not a success as the sentence implies.

Answer: We thank the Reviewer for this point. We have rephrased the sentence specifying that we are referring to the normal survival with therapy compared to other types of malignancy.

Line 63.  It causes Philadelphia chromosome-positive (Ph+) CML 63 and the formation of new fusion genes encoding for the chimeric BCR-ABL1 tyrosine- 64 protein kinase (BCR-ABL1) p210 oncoprotein.

This is oversimplified.  Avoid specifics since there are many different fusion proteins.

Answer: Thank you for this important point. We listed the different BCR::ABL1 isoforms and specified that p210 is the most common in CML patients. Please, see lines 86-87

Line 122. Sokal score, Euro score, and EUTOS score. What about the ELTS?

Answer: Thank you for this point. ELTS score has been added in the text. Please see lines 146 and 151.

Line 505. he clinical case to evaluate. 504 However, the world is not black and white, and it is important, one step before a 505 possible revolution, to be able to identify the pitfalls near the gold nuggets.

Line 584. sneaky

Not standard scientific jargon.

Answer: Thank you for this suggestion. The sentences have been rephrased according to Reviewer’s suggestion. Please, see line 561-563.

This previous reference seems pertinent and not included:

Elhadary M, Elsabagh AA, Ferih K, Elsayed B, Elshoeibi AM, Kaddoura R, Akiki S, Ahmed K, Yassin M. Applications of Machine Learning in Chronic Myeloid Leukemia. Diagnostics (Basel). 2023 Apr 3;13(7):1330. doi: 10.3390/diagnostics13071330. PMID: 37046547; PMCID: PMC10093579.

Answer: Thank you for this important suggestion. The manuscript was included in the Discussion section.

Round 2

Reviewer 4 Report

Comments and Suggestions for Authors

This is  readable article covering an important field.